# DexGraspNet 2.0: Learning Generative Dexterous Grasping in Large-scale Synthetic Cluttered Scenes

**Jialiang Zhang**[1,2,*] **Haoran Liu**[1,2,*] **Danshi Li**[2,*] **Xinqiang Yu**[2,*]
**Haoran Geng**[1,2,3] **Yufei Ding**[1,2] **Jiayi Chen**[1,2] **He Wang**[1,2,4,†]

CFCS, School of Computer Science, Peking University[1]   Galbot[2]
UC Berkeley[3]   Beijing Academy of Artificial Intelligence[4]

**Abstract:** Grasping in cluttered scenes remains highly challenging for dexterous hands due to the scarcity of data. To address this problem, we present a large-scale synthetic benchmark, encompassing 1319 objects, 8270 scenes, and 427 million grasps. Beyond benchmarking, we also propose a novel two-stage grasping method that learns efficiently from data by using a diffusion model that conditions on local geometry. Our proposed generative method outperforms all baselines in simulation experiments. Furthermore, with the aid of test-time-depth restoration, our method demonstrates zero-shot sim-to-real transfer, attaining 90.7% real-world dexterous grasping success rate in cluttered scenes.

**Keywords:** Dexterous Grasping, Synthetic Data, Generative Models

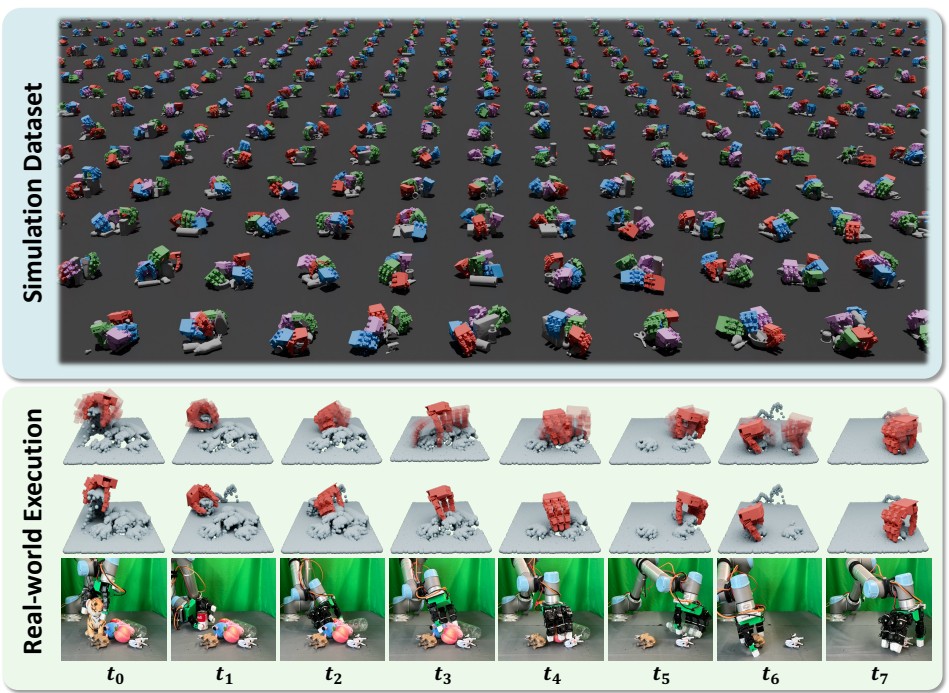

Figure 1: **Overview. Simulation Dataset**: DexGraspNet 2.0 contains 427M grasps (4 random grasps are visualized in each scene here for clarity). **Real-world Execution**: first row are model-generated grasps conditioned on real-world single-view depth point clouds, second row are top-ranked grasps, and third row are real-world executions.

---

*Equal contribution

†Corresponding author: hewang@pku.edu.cn

8th Conference on Robot Learning (CoRL 2024), Munich, Germany.

# 1 Introduction

Recent years have witnessed significant advancements in dexterous grasping datasets [1, 2, 3] and algorithms [4, 5] for single objects. However, extending these advancements to cluttered scenes poses a formidable challenge due to data scarcity. Existing datasets are either too small [6], contain loosely placed objects [6, 7], or rely on naive search methods [6, 8], hindering algorithm development. Furthermore, this slow progress in dataset research obscures the scale requirements of scenes, grasps, and objects needed for effective generalization.

In this study, we present DexGraspNet 2.0, a large-scale synthetic benchmark for robotic dexterous grasping in cluttered scenes. The dataset comprises 8270 scenes and 427 million grasp labels for the LEAP hand [9]. All grasps are synthesized via an optimization process aimed at achieving force closure [10, 1] to ensure diversity and quality.

In addition to data, learning grasping in cluttered scenes presents its own challenges. Firstly, the intricate scene landscapes contribute to a highly complicated distribution of valid grasps, potentially confusing networks. Previous regression methods [8] that directly regress grasp parameters often converge to a mean or median pose in such complex distributions, causing penetration or inaccurate contacts. Secondly, the observation variation of cluttered scenes greatly surpasses that for grasping single objects, demanding higher generalization efficiency. Typical grasping approaches for single objects [11, 12] use the global feature vector to predict grasps, necessitating extensive object-level variations to grasp novel objects. Their direct application in cluttered scenarios could significantly impede generalization to new scenes.

To address these challenges, we propose a method that leverages a generative model conditioned on local features to predict grasp pose distribution. Firstly, employing a generative model allows our method to handle the multimodality of grasp distributions more effectively, enhancing output quality. Secondly, by conditioning on local features, our method better exploits the dataset's diverse variations in local geometries, boosting generalization to new objects and scenes.

To comprehensively evaluate our method, we perform simulation experiments on DexGraspNet 2.0, where our model outperforms all baselines. Additionally, we scale down the dataset to identify the turning point for generalization. Finally, with the aid of test-time depth restoration [13], our model achieves a 90.7% success rate in cluttered dexterous grasping in real-world scenarios, confirming the practicality of our model, which trains on fully synthetic data.

We summarize our contributions as follows: (1) we present DexGraspNet 2.0, a large-scale synthetic dexterous grasping benchmark in cluttered scenes, encompassing 1319 objects, 8270 scenes, and 427 million grasps, (2) we propose a two-stage grasping method that learns efficiently from data by using a diffusion model that conditions on local point features, (3) we conduct systematic analysis and simulation ablations to justify our main design choices, and comprehensive simulation benchmarks along with a 90.7% real-world success rate prove the efficacy of our solution.

# 2 Related Work

## 2.1 Gripper Grasping

Grasping with grippers has been extensively studied, fueled by advances in datasets and sim-to-real techniques. Several works [14, 15, 16] have synthesized large-scale grasping datasets through sampling and filtering. However, these datasets often suffer from visual sim-to-real gaps [15, 16] or require labor-intensive real-world data collection [14]. To address these issues, some works [17, 13] use depth restoration techniques to mitigate visual sim-to-real gaps for point-cloud-based methods.

With these advances, data-driven gripper grasping has achieved significant success. Works such as [14, 18] employ sampling and ranking to predict grasps but face challenges when extending to dexterous hands due to the increased dimensionality. Other approaches [19, 20, 21, 22] directly regress grasp parameters; however, these methods often converge to the mean or median of the data distribution, making them ill-suited for the complex distributions of dexterous grasping. Additionally, while [16, 23] have explored generative gripper grasping, their designs are tailored for grippers. In contrast, our generative method that can be applied to both grippers and dexterous hands.

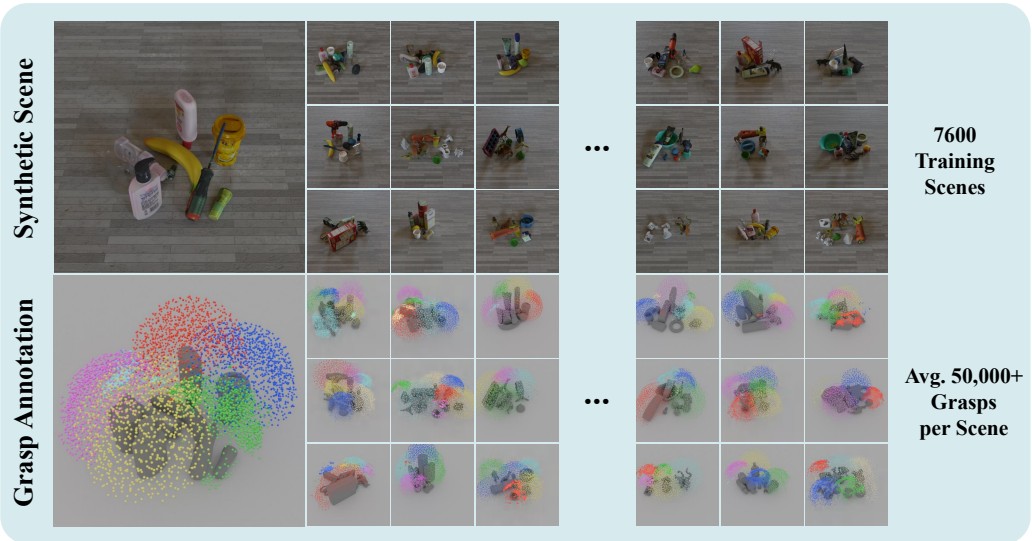

Figure 2: **DexGraspNet 2.0 Benchmark** includes 7600 training scenes and an average of 50k+ grasps per scene, totaling 400M+ grasp labels. The images at the same position in the first and second row correspond to the same scene. Each colored point in the second row represents the palm position of a grasp label, with different colors indicating grasp poses on different objects. For simplicity, grasp labels in each scene are downsampled to 1000.

## 2.2 Dexterous Grasping Datasets

Creating dexterous grasping datasets is more challenging than developing gripper grasping datasets due to higher control dimensions and more complex contact interactions. One line of research [24] employs teleoperation systems to collect such datasets, but struggles with efficiency and scalability. Sampling-based methods [25, 6, 8] can significantly scale up data generation but tend to overly simplify the search space, resulting in limited diversity. Recent works [26, 1, 10, 27, 2] have shifted towards optimization-based methods, which can synthesize stable and diverse grasping poses efficiently by optimizing a designed energy function.

Using these methods, prior studies have generated large grasping datasets for single objects [1, 2]. However, current datasets for cluttered scenes are either too small [6] or too simplistic [7, 8]. Our work introduces the first comprehensive benchmark for dexterous grasping in synthetic cluttered scenes, utilizing optimization-based methods to efficiently generate diverse and high-quality grasps.

## 2.3 Data-driven Dexterous Grasping Methods

Data-driven methods leverage synthetic datasets to learn dexterous grasping from point clouds or depth images. Similar to gripper grasping, research in this area can be categorized into three approaches: sampling/ranking-based, regression-based, and generation-based. Sampling/ranking-based methods [7] often require simplifications and test-time optimizations, which compromise both accuracy and efficiency. Regression-based methods [5, 8, 28] struggle with multi-modal grasp distributions as training data scales up and scenes become more complex. In contrast, generation-based methods, which utilize generative models, effectively learn the complex distribution of diverse grasping poses. Previous works [11, 12] have explored incorporating generative models into dexterous grasping, but not in an end-to-end fashion, and typically focus on single-object scenarios. Our approach uses a diffusion model to learn dexterous grasping in cluttered scenes end-to-end.

Additionally, some studies [29, 4, 30] have explored using reinforcement learning in simulators to train policies for closed-loop dexterous grasping. However, our study focuses primarily on open-loop methods, where a grasping pose is predicted first and then executed.

# 3  DexGraspNet 2.0 Benchmark

We present a comprehensive dexterous grasping benchmark as illustrated in Fig. 2, which is a combination of 1319 diverse objects, 8270 cluttered scenes, and 427M grasps in all scenes.

## 3.1  Object Collection and Scene Synthesis

For training, we generate 7600 synthetic cluttered scenes using all 60 training objects from [14]. Then we collect 1259 unseen objects from [14, 31] and create 670 testing scenes with all 1319 objects. In each scene, 1 to 11 objects are piled within an approximately 30 by 50 cm area, and depth images are rendered from 256 different views. Among the 7600 training scenes, 100 are directly adopted from [14], which are densely packed (containing 8 to 11 objects each). The other 7500 training scenes have a random number of objects. For further details, please refer to our supplementary materials.

## 3.2  Dexterous Grasp Annotation

We employ a two-stage pipeline to annotate dexterous grasp labels within cluttered training scenes. We first synthesize grasp labels for single objects using our modified implementation of [10, 1] and then leverage the IsaacGym simulator [32] to filter out unstable ones (with friction set to 0.2). Then for each scene, we gather grasps from all objects and retain the collision-free ones. We synthesize approximately 1.9M stable grasps for each object (190M in total), resulting in about 56K collision-free grasps for each scene (427M in total). For more details regarding our modifications to [10, 1], please refer to our supplementary materials.

## 3.3  Dexterous Grasp Evaluation in Simulation

We evaluate various models by their success rates in the IsaacGym [32] simulator. For each test scene, a model receives a single-view depth point cloud and produces a grasp pose. If capable of generating multiple grasps, the model must select the best proposal. A grasp is deemed successful if it can lift an object within the simulator. The friction coefficient is set to 0.2, consistent with the dataset's filtering procedure. We design six test groups consisting of densely, randomly, and loosely packed scenes with objects from GraspNet-1Billion [14], as well as these three types of packed scenes with 1231 objects from ShapeNet [31]. For further information, please consult the supplementary materials.

# 4  Generative Dexterous Grasping in Cluttered Scenes

We design a two-stage method to generate dexterous grasp poses in cluttered scenes: (1) a seed point proposal module that identifies graspable regions and extracts point-wise local features, and (2) a grasp pose generation module that models grasp pose distributions conditioned on local features. The combination of the generative model with local conditioning enables our network to learn from numerous local geometry variations in the dataset, which greatly enhances generalization efficiency. We will first introduce the inference process in Sec. 4.1 and Sec. 4.2, and then explain the training process in Sec. 4.3. The entire architecture is demonstrated in Fig. 3.

## 4.1  Seed Point Proposal

Inspired by [18], the seed point proposal module extracts point-wise features $f$ from a single-view depth point cloud $P$ and identifies graspable regions by generating object segmentation score $O$ and graspness score $GS$ for each point. Based on these, we select a subset of high-scoring points, termed seed points, whose local features are fed into the subsequent grasp generation module, achieving more efficient generalization than conditioning on global features [11, 12].

**Ground-truth Graspness Definition.** For each training scene, we define the graspness score $GS$ for every point $p$ on the surface of objects, indicating the level of graspability in its surrounding area. In essence, this score is computed by allowing each grasp label to "vote" for nearby points within reach of the palm, following a heuristic rule. Subsequently, $GS_p$ is defined as the logarithm of the sum of all voted values. Empirically, $GS_p$ reflects the abundance of valid grasps in its vicinity. And these definition details can be found in the supp.

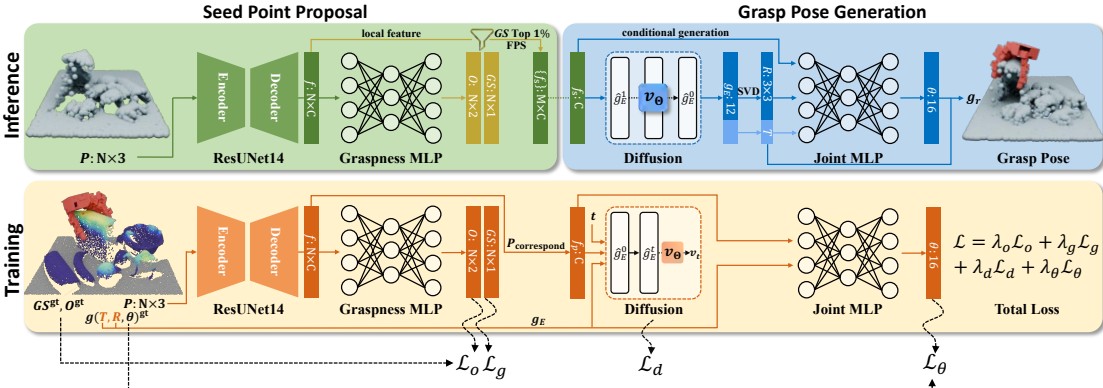

Figure 3: **Method architecture.** Our method leverages a generative model conditioned on local features and models the distribution of grasp poses $(T, R, \theta)$ in a decomposed way. **Inference**: The model receives a single-view depth point cloud and generates multiple grasps (only one is visualized). **Training**: The model takes the depth point cloud and ground-truth annotations to learn data distribution.

**Graspness Prediction and Seed Point Sampling.** To extract local features and predict point-wise graspness, we utilize the same graspness network as described in [18]. Given the scene point cloud $P \in \mathbb{R}^{N \times 3}$, we employ a ResUNet14 built upon MinkowskiEngine to extract a feature vector $f_p$ for each point $p \in \mathbb{R}^3$. This feature is then fed into an MLP to predict $p$'s graspness score $GS_p$ and object segmentation score $O_p$, which classifies whether $p$ belongs to an object or the table. Finally, object points that rank top 1% in $GS$ are selected and downsampled to $M = 1024$ points via FPS (farthest point sampling). These points, denoted as $S = \{s\}$, are termed seed points. Their feature vectors $\{f_s\}$ and graspness scores $\{GS_s\}$ are used for subsequent grasp pose generation.

## 4.2 Grasp Pose Generation

The grasp pose generation module takes a seed point's feature vector $f_s$, and generates diverse dexterous grasp poses relative to that seed point using a generative model. These grasp poses are then ranked based on their estimated log-likelihoods and the graspness $GS_s$ of the seed point.

**Notations and Assumptions.** A dexterous grasp pose relative to a seed point is denoted as $g^r = (T, R, \theta)$, where $T \in \mathbb{R}^3$ and $R \in \mathrm{SO}(3)$ represent the wrist pose relative to the seed point, and $\theta \in \mathbb{R}^{\mathrm{DoF}}$ signifies joint angles of the hand (DoF $= 16$ for LEAP hand [33]). Given a seed point $s$ from a scene point cloud $P$, all valid grasps near $s$ form a conditional probability distribution $p(T, R, \theta | f_s)$, where $f_s$ is the predicted visual feature of point $s$. We assume that the distribution of $(T, R)$ conditioned on $f_s$ is multi-modal and complicated, while the distribution of $\theta$ conditioned on $f_s$ and $(T, R)$ is single-moded. Therefore, we use a conditional generative model to predict the conditional distribution $p(T, R | f_s)$, and a deterministic model to predict $\theta$ from $f_s$ and $(T, R)$.

**Predicting Conditional Pose Distribution via Diffusion.** We adopt the denoising diffusion probabilistic model [34], a powerful class of probabilistic models widely used in the Euclidean space, to approximate $p(T, R | f_s)$. To embed $p(T, R | f_s)$ into the Euclidean space, we flatten the rotation matrix $R$ and concatenate it with the translation $T$ to get the 12D vector representation $g_E$ of a grasp's wrist pose. Then we learn a conditional denoising model $v_\Theta$ to denoise a random 12D Gaussian noise vector $\hat{g}_E^1$ into a valid wrist pose vector $g_E = \hat{g}_E^0$ through an iterative process. Specifically, at each diffusion timestep $t \in [0, 1]$, we feed $t$, feature vector $f_s$, and the current noisy sample $\hat{g}_E^t$ to an MLP $v_\Theta$, which then predicts the velocity [35] of the diffusion process. Using the predicted velocity, we denoise $\hat{g}_E^t$ into $\hat{g}_E^{t-\mathrm{d}t}$ by solving an ODE illustrated in [36]. After the last step, the denoised $g_E = \hat{g}_E^0 \in \mathbb{R}^{12}$ is projected back to $\mathrm{SE}(3)$ by applying SVD [37] to the rotation channels. Moreover, we estimate the sample's probability $p(g_E | f_s)$ by solving a PDE introduced in [38, 36], and then empirically rank the sample with a linear combination of $\log p(g_E | f_s)$ and $GS_s$.

**Joint Angle Prediction.** After sampling a wrist pose $(T, R)$ from the diffusion model, we input $f_s$ and $(T, R)$ into an MLP to predict the joint angles $\theta$, together forming $g^r = (T, R, \theta)$, a generated dexterous grasp pose relative to seed point $s$. Additionally, our method seamlessly extends to parallel grippers by substituting the joint angles $\theta \in \mathbb{R}^{16}$ with one parameter $w \in \mathbb{R}$ indicating gripper width.

| | Method | GraspNet-1Billion | | | ShapeNet | | |
|---|---|---|---|---|---|---|---|
| | | Dense | Random | Loose | Dense | Random | Loose |
| Baseline | HGC-Net [8] | 46.0 | 37.8 | 26.7 | 46.4 | 44.8 | 30.4 |
| | GraspTTA†[11] | 62.5 | 54.1 | 42.8 | 56.6 | 57.8 | 46.4 |
| | ISAGrasp† [5] | 63.4 | 60.7 | 51.4 | 64.0 | 56.3 | 52.7 |
| Ablation | local feature | 16.8 | 10.9 | 4.8 | 21.3 | 17.6 | 10.9 |
| | decomposed model | 84.2 | 80.7 | 71.3 | 74.9 | 72.5 | 66.4 |
| | random scene | 90.0 | **84.1** | 68.2 | 78.9 | 78.8 | 71.3 |
| | Ours | **90.6** | 83.7 | **73.2** | **81.0** | **85.4** | **74.2** |

Table 1: **Baseline and ablation studies for dexterous grasping.** Modified baseline methods are indicated with †. Ablation studies are conducted on three aspects as shown in the lower half of the table. Each **Dense** scene contains 8-11 objects, and each **Random** scene contains 1-10 objects, obtained by deleting objects from Dense scenes, and each **Loose** scene contains 1-2 objects.

### 4.3 Joint Training and Loss Functions

The seed point proposal module and grasp pose generation module are trained jointly. At each gradient step, we randomly sample $D = 8$ scenes from our dataset and select a rendering view for each scene. The depth point cloud of a scene is denoted as $P \in \mathbb{R}^{N \times 3}$, its corresponding object point mask is $\{O_p^{gt}\}_N$, and the ground truth point-wise graspness scores are $\{GS_p^{gt}\}_N$. All point coordinates are represented in the camera frame. We then sample $B = 64$ random grasp labels $\{g\}_B$ in this scene. For each grasp label $g$, we compute its "corresponding point" $p$ following Section 4.1 and represent $g$ as a relative grasp pose $(T^{gt}, R^{gt}, \theta^{gt})$ in the reference frame of $p$.

First, point cloud $P$ is fed into the seed point proposal module to obtain local features $\{f_p\}_N$, object segmentation scores $\{O_{p,0/1}\}_N$, and point-wise graspness scores $\{GS_p\}_N$, after which the object segmentation loss $\mathcal{L}_o$ (Cross-Entropy) and graspness loss $\mathcal{L}_g$ (SmoothL1) are computed.

Next, for each grasp $g_j$, we collect the local feature $f_p$ of its corresponding point $p$ and pass these to the grasp generation module. The 12D Euclidean representation of the wrist pose $g_E$ undergoes the diffusion process to obtain a noisy sample $\hat{g}_E^t = \sqrt{\overline{\alpha}_t}g_E + \sqrt{1 - \overline{\alpha}_t}\epsilon$ and diffusion velocity $v_t = \sqrt{\overline{\alpha}_t}\epsilon - \sqrt{1 - \overline{\alpha}_t}g_E$ at some random time step $t$. The denoising model $v_\Theta$ then predicts this velocity, under the supervision of an MSE loss $\mathcal{L}_d$. The joint angle prediction MLP takes feature $f_p$ and the wrist pose $(T^{gt}, R^{gt})$, predicts $\theta$, and is supervised by a joint angle loss $\mathcal{L}_\theta$ (Smooth L1).

The total loss is a linear combination of all loss terms: $\mathcal{L} = \lambda_o\mathcal{L}_o + \lambda_g\mathcal{L}_g + \lambda_d\mathcal{L}_d + \lambda_\theta\mathcal{L}_\theta$. The model is trained on a single NVIDIA 3090 for 50k iterations, taking approximately 2 hours.

## 5 Experiment

We compare our method with several representative approaches and conduct ablation studies on DexGraspNet 2.0. Additionally, we scale down our dataset and compare the turning point for the generalization of different models. We also evaluate our method on gripper grasping. Finally, we conduct real-world experiments to confirm the practicality of our approach.

### 5.1 Experiments on DexGraspNet 2.0

**Baseline comparisons.** We compare our method with HGC-Net [8], GraspTTA [11], and IS-AGrasp [5] on DexGraspNet 2.0, with results in Tab. 1. Note that GraspTTA [11] and ISAGrasp [5] were originally designed for single-object grasping, so we modify them for the cluttered setting, denoting these versions as GraspTTA† and ISAGrasp† (details in the supp). Both HGC-Net [8] and ISAGrasp† [5] use regression approaches, which struggle with complex distributions and yield subpar performance. Although GraspTTA†[11] incorporates a CVAE to predict grasp distributions, it still suffers from low network capacity. In contrast, our method, which utilizes a diffusion model to better capture complex grasp distributions, significantly outperforms all baseline methods.

**Ablation Studies.** Our method aims to model the complex distribution of dexterous grasping while achieving higher generalization efficiency by leveraging a generative model that conditions on local features. To better analyze the effectiveness of each module, we conduct ablation studies on three aspects: (1) local feature (2) decomposed pose modeling (3) randomly-packed training scenes. As shown in Tab. 1, substituting local feature conditioning with global ones yields poor performance.

To illustrate, global conditioning relies on scene-level variations to generalize, which are limited in number, whereas point-wise local features derive numerous diverse geometry patches (with paired grasp labels), greatly enhancing generalization efficiency. Moreover, replacing decomposed pose modeling with the combined modeling of $(R, T, \theta)$ also leads to a perceivable decline in performance due to the incompatibility of the Euclidean space with SO(3) space. Finally, our dataset design that incorporates scenes with random object numbers proves to be effective.

## 5.2 Scaling the Dataset

We scale down the training data in two ways: (1) by decreasing the number of training scenes, and (2) by reducing the number of grasps in each scene. The success rate is evaluated across the two densely packed test groups and averaged, as shown in Fig. 4.

**Scenes.** We study the impact of scene numbers on performance. Our model consistently improves with more training scenes. Notably, despite being trained on only 60 objects, it generalizes well to test scenes containing 1231 novel objects, which suggests that scene diversity matters more than object variety. This finding is encouraging, as it indicates that grasp labels from individual objects can be reused to compose new scenes, enabling performance gains with minimal additional effort.

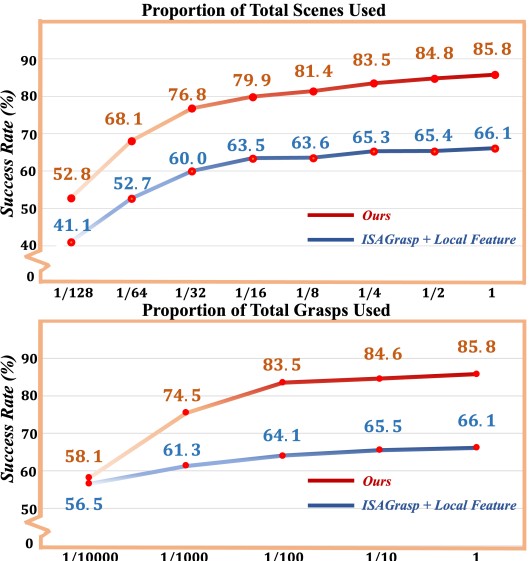

Figure 4: Scaling the number of scenes/grasps.

**Grasps.** We investigate how generative and regressive methods scale with increased training grasps. Our model shows significant performance gains as the grasp dataset grows, highlighting its ability to scale effectively with more grasps. In contrast, the success rate of GraspTTA[†] increases only marginally before quickly plateauing, suggesting that regression-based methods struggle to fully leverage larger grasp datasets due to their inability to handle complex data distributions.

| Method | AP Seen | AP Similar | AP Novel |
|---|---|---|---|
| GG-CNN [39] | 15.5/16.9 | 13.3/15.1 | 5.5/7.4 |
| Chu *et al.* [40] | 16.0/17.6 | 15.4/17.4 | 7.6/8.0 |
| GPD [41] | 22.9/24.4 | 21.3/23.2 | 8.2/9.6 |
| PointGPD [42] | 26.0/27.6 | 22.7/24.4 | 9.2/10.7 |
| GraspNet1B[14] | 27.6/29.9 | 26.1/27.8 | 10.6/11.5 |
| GSNet [18] | 67.1/63.5 | 54.8/49.2 | 24.3/19.8 |
| GSNet [18]* | **68.7** | 59.8 | 24.6 |
| Ours* | 56.0 | 53.2 | 23.2 |
| Ours # * | 66.6/61.5 | **61.7**/53.3 | **27.4**/23.2 |
| GSNet (render)* | **78.4** | 69.7 | 36.7 |
| Ours (render) # * | 77.4 | **72.5** | **39.1** |

Table 2: **Experiment results on GraspNet-1Billion.** * means using grasps that scores over $\geq 0.9$. # means using refined poses, and render means using rendered depth images. Statistics in the table follow a **RealSense/Kinect** format, where results with a single number use the Realsense setting.

## 5.3 Simulation Experiments on Gripper Grasping

We evaluate our method's performance in gripper grasping using the widely adopted benchmark GraspNet-1Billion [14]. This benchmark categorizes test scenes based on seen/similar/novel objects

and evaluates methods using the average precision (AP) metric. Additionally, we take three steps to refine its training data: (1) retain only grasps with scores of $\geq 0.9$, (2) refine the grasps, and (3) use the ground-truth depth instead of real-world depth for training and evaluation (applying depth restoration [13] for subsequent real-world experiments). These refinements reduce training grasps from 1B to 4.2M. The results, shown in Tab. 2, indicate that our method performs comparably to GSNet [18] when trained on the same data. This finding, along with the results in Sec. 5.1, suggests that our method serves as a unified framework for both dexterous and gripper grasping. Furthermore, both our method and GSNet [18] achieve higher scores with the refined training data, highlighting that data quality is more important than quantity for gripper grasping.

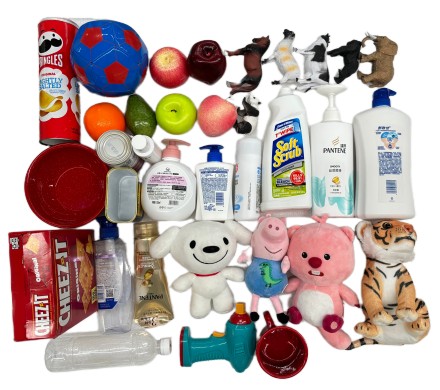

Figure 5: Real-world experiment objects.

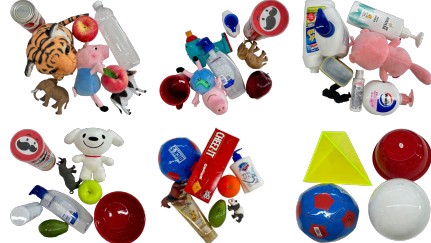

Figure 6: Real-world experiment scenes.

| Method | SR (%) | | Method | SR (%) |
|---|---|---|---|---|
| HGCNet | 16.4 | | | |
| GraspTTA[†] | 6.0 | | ASGrasp | 84.5 |
| ISAGrasp[†] | 54.8 | | Ours | **92.5** |
| Ours | **90.7** | | | |

Table 3: Success Rate (SR) of real-world experiments. Left: dexterous hand. Right: gripper.

## 5.4 Real-World Experiments

We conduct real-world experiments for both dexterous and gripper grasping. The dexterous grasping experiments utilize a LEAP hand [33] mounted on a UR-5 robotic arm, while the gripper experiments employ a Franka Panda arm. In both setups, an Intel RealSense D435 camera captures a single depth map from a top-down perspective. Since depth sensors often struggle with transparent or specular objects, we apply an off-the-shelf depth restoration method [13] to preprocess the depth maps before generating the depth point clouds. This restoration step is ablated in the supp.

We collect 32 objects and compose 6 test scenes as illustrated in Fig. 5 and Fig. 6. The grasping system iteratively removes objects until the table is cleared or until two consecutive failures occur. The results, presented in Tab. 3, demonstrate that our method achieves a successful grasp rate of 90.7% with the dexterous hand and 92.5% with the gripper, surpassing baseline methods. Additionally, on an NVIDIA 4090 GPU, seed point proposal and grasp generation together take less than 0.5 seconds, with depth restoration adding only 0.2 seconds.

## 6 Limitations and Conclusion

In this paper, we present DexGraspNet 2.0, a large-scale benchmark for dexterous grasping in cluttered scenes. Our proposed method, which leverages a generative model conditioned on local features, outperforms all baselines in simulation and achieves a 90.7% success rate in real-world tests. We believe that our insights into grasping methodologies and dataset construction could offer valuable guidance for researchers working on similar tasks.

Nonetheless, our work has several limitations. First, our method is not suitable for dynamic scenes since the framework is open-looped. Second, following up on the first point, our grasping policy is a simple heuristical squeeze-and-lift motion. Therefore, it struggles with scenarios that require more complicated and functional actions, like flipping a thin and low object off the table. Third, because the system has no tactile feedback, it is difficult to handle scenes with heavy occlusions. Fourth, our method has trouble picking up very small objects, which may be because the fingers of the LEAP hand are too thick. We look forward to future works that address these limitations.

**Acknowledgments**

This work was supported by PKU-Galbot Joint Lab of Embodied AI.

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

## Supplementary Material

This document provides supplementary details, additional experiments, and enhanced visualizations to complement the main paper. Sec. A outlines the detailed experimental settings discussed in the main paper. Sec. B presents additional experiments conducted to extend the findings. Sec. C offers further statistical insights into our DexGraspNet 2.0 benchmark. Sec. D details the methodology used for generating grasp labels. Sec. E elaborates on the technical aspects involved in constructing and training our model for dexterous grasping in cluttered scenes. Sec. F highlights specific implementation details related to applying our model with parallel grippers. Sec. G provides additional visualizations showcasing our dataset and model.

## A    Experiment Details

We provide additional details on the experiment settings due to space constraints in the main paper. Sec. A.1 delineates how we evaluate a grasp in a simulator and enumerates some of the physics parameters involved. Sec. A.2 elaborates on the three ablation groups in detail. Sec. A.3 outlines the three baseline methods benchmarked in the main paper.

### A.1    Evaluation Metric

We evaluate various grasping models by measuring their simulation success rates in the Isaac Gym simulator. For each test scene, a model is expected to take a single-view depth point cloud as input and output one grasp pose $G_p$. If capable of generating multiple grasps, the model must select the best proposal, as required in the main paper. Following this, the evaluator determines whether $G_p$ constitutes a successful grasp. Specifically, a predefined rule is applied to calculate a pregrasp pose, squeeze pose, and lift pose, thereby establishing a complete action trajectory $T$. Subsequently, $T$ is executed within the simulator, and success is determined by its ability to lift an object off the table without any initial intersection with the table or surrounding objects. Consistency is ensured across all experiments by maintaining the same trajectory generation rule and physics parameters. Some of the important physics parameters are listed in Tab. 4.

| Parameter | Value | Parameter | Value |
|---|---|---|---|
| friction coeff | 0.2 | object mass | 0.1 kg |
| joint stiffness | 800 | joint damping | 20 |

Table 4: Physics Parameters

### A.2    Ablation Details

We explain the settings of the three ablation groups from our main paper in detail.

**Local Feature.** Our grasping method aims to achieve higher generalization efficiency by conditioning on local features. We investigate this design by training a diffusion model that predicts the distribution of all valid grasps conditioned on the scene's global feature. This ablated version has three major differences compared to our original model: (1) it discards the UNet decoder and retains only the encoder; (2) during training, each grasp label corresponds to the global feature vector of the scene point cloud (output by the encoder) instead of the local feature vector of the grasp's corresponding point (one of the point-wise vectors output by the decoder). (3) during inference, the model does not predict graspness or propose seed points, but only encodes the scene point cloud and directly generates grasp poses conditioned on its global feature vector.

**Decomposed Pose Modeling.** Our grasp generation module models the conditional distribution $p(T, R, \theta|f_s)$ in a decomposed manner: a conditional generative model predicts the conditional distribution $p(T, R|f_s)$, followed by a deterministic model predicting $\theta$ from $f_s$ and $(T, R)$. Surprisingly, the above design slightly outperforms a seemingly more elegant approach: using a single conditional generative model to fit the joint distribution $p(T, R, \theta|f_s)$ without decomposing the wrist

pose $(T, R)$ from the joint angles $\theta$. We postulate that this phenomenon results from the distribution of the training data, rather than an inability to properly tune the second approach. Specifically, our training dataset primarily consists of power grasps that utilize all fingers, resulting in a single-mode distribution of $\theta$ conditioned on $(T, R)$ and $f_s$. Consequently, the deterministic model regressing $\theta$ is not confused by this data distribution; instead, it potentially becomes more robust to outliers. Essentially, the outcomes of this ablation group are highly specific to our task and training data. If we incorporate additional grasping modes into our dataset, such as precision grasps and functional grasps, it would violate our assumption of a single-mode distribution of $\theta$ conditioned on $(T, R)$ and $f_s$. In such a scenario, jointly modeling $p(T, R, \theta | f_s)$ with a single conditional generative model might outperform our current design.

**Randomly-Packed Training Scenes.** In addition to ablating our network designs, we also conduct one experiment to ablate our dataset in the main paper. Our training set comprises 100 densely packed scenes (with 8 to 11 objects) and 7500 randomly packed scenes (with 1 to 10 objects). All dense scenes are sourced from [14]. However, we observed that training solely on these dense scenes resulted in the inability to generate valid grasp poses when the table is nearly clear. Therefore, we incorporated the randomly packed scenes to ensure performance across all density levels.

### A.3 Baseline Details

We outline the three baselines compared in the main paper and detail how we adapted two of them from their original setting of single-object grasping to our cluttered scenarios.

**HGC-Net [8].** HGC-Net is a two-stage method for grasping in cluttered scenes. Initially, a segmentation model divides the scene point cloud into graspable points and ungraspable points. Following this, a deterministic model predicts a grasp pose near each graspable point. Given that this method already focuses on cluttered scenes, minimal modifications were required. The only change made was switching their end effector from the HIT-DLR II hand to the LEAP hand.

**ISAGrasp [5].** ISAGrasp is a regressive method designed for grasping single objects. It employs a PointNet++ encoder [43] to encode the object point cloud into a global feature vector. Subsequently, an MLP is utilized to predict the wrist translation, wrist quaternions, and joint angles. We extensively modified this method to adapt it for cluttered scenes: (1) We replaced their PointNet++ encoder with a ResUNet14 encoder-decoder and incorporated a seed point proposal module based on point-wise graspness prediction, similar to our method. (2) During inference, this modified model predicts the grasp parameters from the local feature vector of the proposed seed point, instead of the global feature vector obtained from their original point cloud encoder. (3) During training, each grasp label is associated with its corresponding point rather than its target object. We designate the modified model as ISAGrasp$^\dagger$. It is worth noting that this adaptation already rectifies a major suboptimal aspect of their original baseline by integrating one of our key designs: replacing global conditioning with local conditioning. Consequently, the adapted method differs from our model solely in the use of a regressive model to predict the wrist pose, whereas we employ a conditional generative model.

**GraspTTA [11].** GraspTTA utilizes a CVAE for grasping single objects. It leverages PointNet [44] to encode the object point cloud into a global feature vector, which serves as conditioning for the CVAE to predict the distribution of the wrist translation, wrist axis angles, and joint angles. We adapt it for cluttered scenes using the same approach as ISAGrasp$^\dagger$, and denote the adapted version as GraspTTA$^\dagger$. Furthermore, we discard the test-time optimization of the original method because it relies on the full point cloud, which is an invalid assumption in our task settings.

## B  Additional Experiments

### B.1  Ablate Rotation Representation

Our method employs the rotation matrix to represent wrist rotation and applies SVD [37] to orthogonalize network predictions. We compared this design against several alternatives: **Euler Angle**

| | Method | GraspNet-1Billion | | | ShapeNet | | |
|---|---|---|---|---|---|---|---|
| | | Dense | Random | Loose | Dense | Random | Loose |
| Ablation | Euler Angle | 87.6 | 82.0 | 73.0 | 78.0 | 76.4 | **75.2** |
| | Axis Angle | 86.4 | 81.7 | 70.5 | 79.0 | 76.4 | 74.1 |
| | Quaternion | 87.9 | 81.5 | 72.0 | 78.6 | 77.0 | 72.9 |
| | 6D | 88.2 | 81.5 | 71.9 | 80.2 | 79.0 | 73.0 |
| | Ours | **90.6** | **83.7** | **73.2** | **81.0** | **85.4** | 74.2 |

Table 5: **Ablation studies for representations of rotation. Euler Angle** represents rotation as 3D Euler angle; **Axis Angle** represents rotation in 3D as the angle of rotation multiplies the rotation axis; **Quaternion** represents rotation as 4D quaternion; **6D** represents rotation with the first two rows of the rotation matrix. **Ours** represents the rotation as the rotation matrix.

| | Method | GraspNet-1Billion | | | ShapeNet | | |
|---|---|---|---|---|---|---|---|
| | | Dense | Random | Loose | Dense | Random | Loose |
| Ablation | Graspness | 81.8 | 76.6 | 68.0 | 73.7 | 71.3 | 64.4 |
| | Log Probability | 78.1 | 78.4 | 75.1 | 72.4 | 71.6 | **74.6** |
| | Random | 65.1 | 62.0 | 57.2 | 61.7 | 58.9 | 56.4 |
| | Ours | **90.6** | **83.7** | **73.2** | **81.0** | **85.4** | 74.2 |

Table 6: **Ablation studies for sampling strategy. Graspness** ranks samples by graspness score only; **Log Probability** ranks samples by log probability only; **Random** randomly draws from sampled poses; **Ours** ranks samples by combination of graspness scores and log probabilities.

(representing rotation as 3D Euler angles), **Axis Angle** (rotation represented by the angle of rotation multiplied by the rotation axis), **Quaternion** (represented as a 4D quaternion), and **6D** (using the first two rows of the rotation matrix). The results in Tab. 5 demonstrate that our choice outperforms all other methods across the evaluated task. As discussed in [37], rotation representations in Euclidean space with fewer than five dimensions, such as Euler angles, axis-angle, and quaternions, are inherently discontinuous. Although the 6D representation circumvents this issue, it is coordinate-dependent. Introducing small noises in different directions to the rotation in a 6D representation results in changes of varying magnitudes. In contrast, our 9D representation is both continuous and coordinate-independent, thereby outperforming other rotation representations.

## B.2 Ablate Ranking Strategy

During inference, we rank all predicted samples to identify the best one using a linear combination of the graspness scores of the seed points and the estimated log probabilities of the wrist poses. We ablate this ranking strategy by removing the graspness score, the log probability, or both. Tab. 6 presents the results. Our method (**Ours**), which ranks samples based on a combination of graspness scores and log probabilities, consistently outperforms the other strategies. Ranking solely by graspness scores (**Graspness**) or log probabilities (**Log Probability**) yields moderate performances, while selecting samples randomly (**Random**) results in the lowest success rates. These findings underscore the efficacy of our proposed ranking strategy in identifying optimal grasp poses.

Interestingly, despite the theoretical challenges in defining a probability density function $p(T, R|f_s)$ on a 6-dimensional data manifold embedded within a higher-dimensional parameter space (12D), experiments demonstrate that our estimated log probabilities consistently enhance the performance of our ranking strategy. Nevertheless, we acknowledge this theoretical inelegance and defer the solution to future studies, such as exploring the use of normalizing flows on $SE(3)$ or employing manifold diffusion methods.

## B.3 Scaling the Dataset for Grippers

We scale down the training data of the parallel gripper by (1) reducing the number of grasps in each scene, and (2) decreasing the number of training scenes. We evaluate the AP metric in simulation for each setting and success rate in the real world.

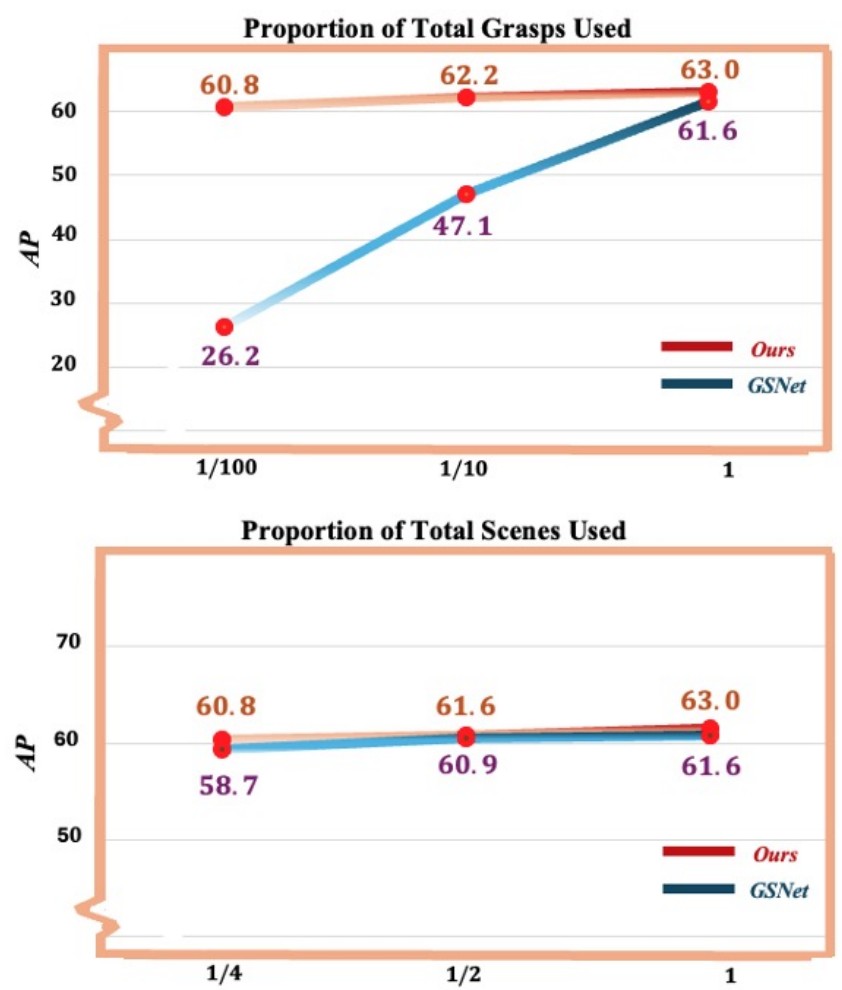

Figure 7: **AP metric** evaluated on models trained with downscaled dataset. **Top**: downscaling the number of grasp labels in each scene. **Bottom**: downscaling number of scenes trained on

| Fraction of Grasps | Success Rate |
|---|---|
| 1/100(42k) | 81.3 |
| 1(4.2M) | 92.4 |

Table 7: **Success Rate of real-world experiment on Ours model trained over downscaled dataset**. We train Ours model with random 1/100 fraction of grasp labels and the entire grasp pose dataset, amounting 42k and 4.2M labels, respectively.

As shown in Fig. 7, although under the full-data setting our generative model only slightly outperforms GSNet by +1.4 AP, the AP metric of GSNet drops by a significant amount of 35.4 as we downscale the number of grasps by 100, whereas our generative pipeline drops by only 2.2. This suggests that our generative pipeline is significantly more sample-efficient than GSNet. Both methods are robust to downscaling of number of training scenes at the scope of our experiment, with only a slightly dropped AP.

The resulting statistics in terms of AP are much to our surprise, as being trained with 1/100 total grasp labels, namely only 42k grasp labels, our generative model seems to still retain strong performance. In order to validate this counter-intuitive result, we carry out real-robot experiments with Ours models trained with a downscaled number of grasps and report success rate in Tab. 7. With 42k

| Depth Restoration | Diffuse | Trans | Hybrid |
|---|---|---|---|
| With | 94.1 | 80.0 | 90.7 |
| Without | 94.1 | 50.0 | 86.4 |

Table 8: Real-world cluttered scene dexterous grasping with/without depth restoration. **Diffuse** includes only diffuse objects, **Trans** comprises only transparent or specular objects, and **Hybrid** includes scenes used in the main paper, consisting of a mixture of diffuse, transparent, and specular objects for comparison.

| End Effector | Normal | Large |
|---|---|---|
| Parallel Gripper | 92.4 | 0.0 |
| Dexterous Hand | 81.5 | 100.0 |

Table 9: Comparison of real-world grasping performance using a parallel gripper or a dexterous hand across different scene types. The five **Normal** scenes consist of typical cluttered environments, while the **Large** scene includes 4 large objects.

training labels, our generative model achieves an 81.5% success rate in real-world cluttered scenes as shown in Fig. 8, which is affirmative to the AP statistics.

In summary, the experiments in this section give strong evidence that the distribution of valid grasp poses does exist and the amount of data required to simulate at least a valid support of such a distribution may prove to be much smaller than previously been conjectured.

### B.4 Using Raw Depth in the Real World

In our real-world experiments, we integrated depth restoration techniques [13] to facilitate grasping transparent and specular objects amidst cluttered scenes. Here, we conduct additional experiments to demonstrate that our method do not rely on depth restoration when grasping diffuse objects. We constructed four additional cluttered scenes in the real world: two scenes (**Diffuse**, as shown in Fig. 8) consisting solely of diffuse objects and two scenes (**Trans**, as shown in Fig. 9) containing only transparent and specular objects. The original five test scenes from the main paper, which include a mixture of objects, are denoted as **Hybrid**. We then evaluated our model on all test groups both with and without the application of depth restoration techniques. The results in Tab. 8 demonstrate two key findings: firstly, our model's effectiveness in real-world grasping is independent of depth restoration for **Diffuse** scenes; secondly, our model exhibits enhanced robustness to object texture, particularly transparent and specular surfaces when depth restoration is applied.

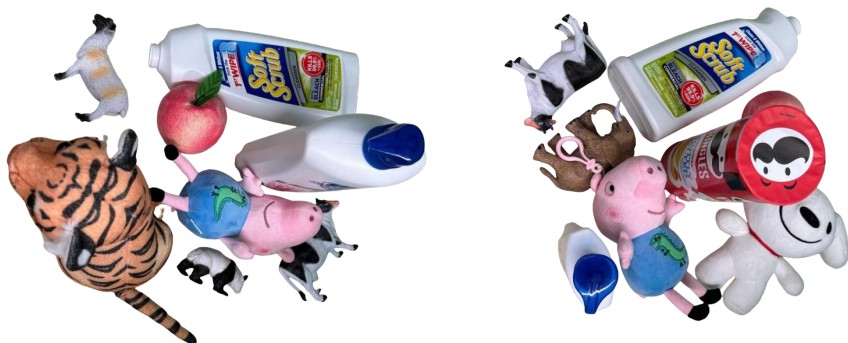

Figure 8: Two **Diffuse** scenes in real world.

### B.5 Discussion on Dexterous Hands vs Parallel Grippers

While grasping systems utilizing parallel grippers have already achieved impressive robustness in the real world [18, 45], we advocate that dexterous hands can further enhance performance. In

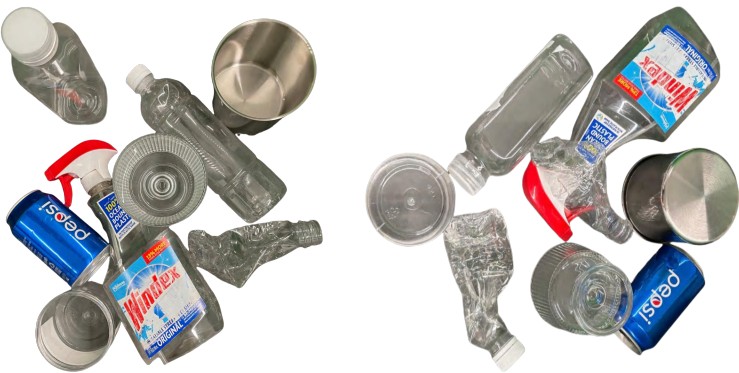

Figure 9: Two **Trans** scenes in real world.

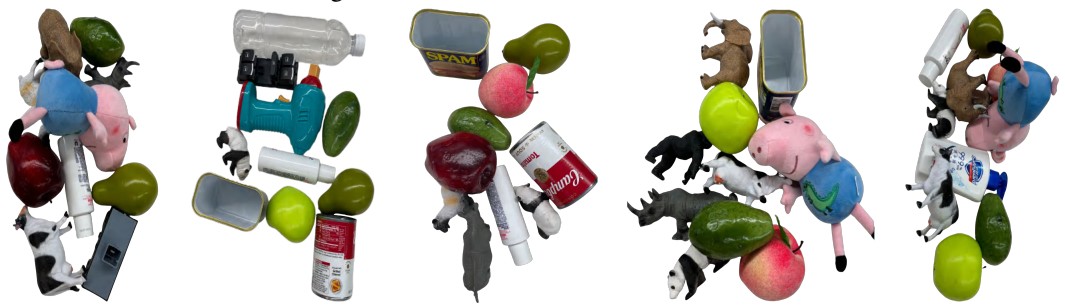

Figure 10: Five **Normal** test scenes for gripper in the main paper.

addition to the 5 test scenes (**Normal**, as shown in Fig. 10) demonstrated in the main paper, we also construct an additional scene (**Large**) consisting of 4 large objects, as shown in the main paper. Real-world experiment results in Tab. 9 indicate that the dexterous hand can grasp each object in this scene, whereas the parallel gripper cannot grasp any object. This is because the dexterous hand possesses strong envelopment capabilities, allowing it to grasp larger objects effectively.

## C   Benckmark Specifications

This Section presents further details about the DexGraspNet 2.0 benchmark proposed by this work. Sec. C.1 provides statistics of the DexGraspNet 2.0 benchmark, including both the **Training Set** that contains ground truth grasp pose annotations and the **Test Set** with no ground truth provided. Sec. C.2 identifies the objects used to generate our benchmark. Sec. C.3 presents the pipeline used to generate training scenes with selected objects. Sec. C.4 elaborates on the protocol of generating test scenes and how we divide them into different splits.

### C.1   Benchmark Statistics

| Splits | number of objects | number of scenes |
|--------|-------------------|------------------|
| Training | 60(GraspNet1B) | 100(seminal)+7500(augmented) |
| Test | 88(GraspNet1B) + 1231(ShapeNet) | 670 |
| Total | 88(GraspNet1B) + 1231(ShapeNet) | 8270 |

Table 10: Statistics of the DexGraspNet 2.0 Benchmark

Tab.10 illustrates the overall statistics. The entire benchmark encompasses two components: a **Training Set** used to train our models and a **Test Set** to evaluate dexterous grasping pose generation models on. Note that ground truth grasp pose annotations are only provided for the training set. In total, the benchmark contains 8270 scenes, 1319 objects, and 426.6M grasp pose annotations.

**Training Set** contains 7600 scenes and 60 objects in total. all training objects are from the GraspNet-1Billion [14] dataset

**Test Set** contains 670 scenes and 1319 objects in total. the 88 objects from the GraspNet-1Billion [14] dataset are used to compose 450 of the test scenes and 1231 objects picked from ShapeNet [31] are used to compose the remaining 220 test scenes

## C.2 Object Selection

The 60 objects in the Training Set are those that appeared in GraspNet-1Billion [14] scenes 0000-0099. The Test Set contains 1319 objects, 88 of them are all the objects in GraspNet-1Billion [14], and the remaining 1231 objects are picked from ShapeNetSem [31].

## C.3 Training Scenes Specification

In the 7600 training scenes, 100 are called **seminal scenes**, which corresponds to the Scenes 0000-0099 in the GraspNet-1Billion [14] dataset composed and rendered using their official meshes and annotations. We augment each seminal scene 75 times by randomly deleting objects in the scene. In each augmented scene, the number of objects deleted is uniformly sampled from [1,$k$-1], where $k$ is the number of objects in the original scene. In total, we generate 7500 augmented training scenes with 100 seminal scenes, totaling 7600 scenes in the entire training set.

## C.4 Test Set Scenes Specification

As shown in Tab. 1 of the main paper, the Test Set is divided into 6 splits. In the following, we specify each of these splits.

**GraspNet-1Billion Dense** composes of 90 scenes that correspond to the Scenes 0100-0189 in the GraspNet-1Billion [14] dataset. Each scene contains 8-11 objects.

**GraspNet-1Billion Random** composes of 180 scenes. This split is generated by augmenting each GraspNet-1Billion Dense split scene twice with the process as described in Sec.C.3

**GraspNet-1Billion Loose** composed of 180 scenes by augmenting each GraspNet-1Billion Dense split scenes twice with the process as described in Sec.C.3, with only 1-2 random objects remaining in the scene.

The three ShapeNet splits are generated by dropping objects on a 30cm×50cm tabletop. In specific, we follow the scene generation process of DREDS [46] with the material randomization function disabled. We run the scene generation process in PyBullet [47] and filter physically stable ones in IsaacGym [32]. The Dense/Random/Loose splits are divided according to the number of objects appearing in each scene.

**ShapeNet Dense** composes of 100 scenes, each containing 8-11 objects

**ShapeNet Random** composes of 90 scenes, each containing 5-9 objects

**ShapeNet Loose** composes of 30 scenes, each containing 1-2 objects

# D Grasp Label Generation

This section elaborates on our pipeline for generating dexterous grasping poses on single objects. First, we define initial hand poses by retargeting GraspNet-1Billion [14] annotations to dexterous hands. Then we run physics-based optimization to generate stable grasps. To maximally diversify the produced data, we adopt two different methods, [10] which targets Grasp Wrench Space (GWS) optimality, and [1] which targets force-closure, as optimization algorithms, each generating half of the dataset. Lastly, we filter stable and collision-free grasps via simulation in the IsaacGym [32] simulator. As shown in Fig. 11, in total we generated 44.9M stable grasp poses for 88 objects from 280M initial poses. Even in the face of our very strict friction coefficient $\mu$=0.2, our method still maintains an overall success rate of 16.07%. In the following subsections, we detail each of these components.

### D.1 Hand Pose Initialization

As discovered in [1], the success rate of dexterous grasp generation is very sensitive to the initial hand pose. Moreover, we aim to cover valid grasp modes for each object as comprehensively as possible. Therefore, we initialize dexterous hand poses by retargeting the exhaustive GraspNet-1Billion [14] gripper annotations.

In specific, we filter points where stable gripper grasp poses are annotated in [14] as grasp points. As shown in Fig. 13, for each grasp point, we align the +y axis (pointing forward out of the palm) of dexterous hand with the +x axis of gripper pose annotation, retreat the center of palm a fixed distance from grasp point in the approaching direction, initialize hand joint qpos with a set of predefined values and exhaustively apply transformations corresponding to 256 approaching directions, 4 depths and 12 in-plane angles as defined in [14].

### D.2 Grasp Pose Optimization

#### D.2.1 GWS-based optimization (adapted version of [10])

We reimplement [10] on the CuRobo [48] framework for better computation parallelism. We set the target Task Wrench Space (TWS) as a unit sphere in 6D wrench space such that the task objective is identical to forming a force-closure grasp and running 600 iterations with naive gradient descent.

#### D.2.2 force-closure-based optimization (adapted version of [1])

We adopt [1] with modification in its definition of force-closure energy, and reimplement the modified algorithm on the CuRobo [48] framework as well.

We observe that the force-closure energy used in [1] assumes unit contact force is applied to each contact point, whereas humans naturally adjust contact forces applied to different contact points in order to maintain a firm grasp. The above assumption limits the objective of optimization in [1] onto a submanifold of the space of all valid grasp poses, hurting the quality and diversity of generated data. Following the notations in [1], we relax the unit-contact force assumption by reformulating the force closure energy as the following bilevel form:

• At each timestep, given the current hand pose, we solve the optimal contact forces applied to current contact points such that the total wrench imposed on the object is minimized. We formulate this intuition into the following linear program:

$$P_t = \min_{\lambda_t} \|G(\lambda_t \odot c)\|_2$$
$$s.t. \max_i (\lambda_t)_i = 1$$
$$(\lambda_t)_i \geq 0, i = 1, 2, ..., n$$

Where $P_t$ has the physical meaning as the total wrench applied to the object when the combination of contact force magnitude, $\lambda_t$, is applied to the contact points. $\odot$ means element-wise product. Note this linear program admits a closed-form solution and therefore imposes a neglectable computation burden.

• Across timesteps, we optimize the differentiable force-closure metric in awareness of the plausibility of the current hand pose:

$$E_{FC} = \begin{cases} \|G(\lambda_t \odot c)\|_2, & \text{if } P_t < \tau_{FC}, \min_i (\lambda_t)_i \geq \tau_\lambda, \text{and } B = 1 \\ \|Gc\|_2, & \text{otherwise} \end{cases}$$

Where $\tau_{FC}, \tau_\lambda$ are predefined thresholds, and $B$ is a binary random variable with $P(B = 1) = 0.9$.

If the current hand pose is already capable of forming a force-closure grasp on the object, mathematically defined as $P_t < \tau_{FC}$ (total wrench acceptably small) and $\min_i (\lambda_t)_i \geq \tau_\lambda$ (a minimum contact

force is applied to each contact point), then we decide the current pose is good enough in terms of force-closure property. In this case, we scale the force closure energy to prevent overoptimization. In effect, the force closure energy now works as a regularization term. Otherwise, if the current hand pose is not stable enough, we keep searching for more stable poses by optimizing the force closure metric with the original energy term. In addition, even for the former case, we stochastically use the original energy term with probability 0.1 to encourage forming more robust grasp poses.

Note in the above formulation, the global minimum set of hand poses for $E_{FC}$ are the poses for which there exists a non-trivial contact force combination such that the total wrench executed to the object is zero. This global minimum set exactly corresponds to the original definition of force closure in [49].

### D.3 Filtering Stable and Collision-Free grasps

We perform grasp filtering in the IsaacGym simulator. First, we check for each grasp pose if the penetration between the hand mesh and object mesh is below 2 mm. For all collision-free grasps, we execute the grasp with a predefined heuristic and simulate for 60 timesteps at 60Hz. The grasp pose is validated as stable if it can deny gravity in all 6 axis-aligned directions. The friction coefficient $\mu$ for both hands and objects are set to 0.2, making the filtering process very strict.

Fig. 11 shows the **Valid Rate** for each object, which is defined as the portion of generated grasps that are both collision-free and stable. The overall success rate is 16.07%, as we generated in total 44.9M valid grasp poses out of 280M grasp pose initializations. The method-specific valid rate for [10] and [1] are 7.91% and 24.19% respectively.

## E   Implementation Details for Dexterous Hands

In this section, we elaborate on the data organization (Sec. E.1) and model architecture (Sec. E.2) of our method for dexterous grasping.

### E.1   Data

**Data Reblancing** In each training scene, the number of grasp labels on graspable objects may be uneven. Randomly sampling grasp labels uniformly across all valid ones in each scene could slow down the learning of grasping objects that have fewer labels. To address this, we implement a two-stage sampling approach to rebalance the training process: first, we randomly sample a graspable object, and then we randomly sample one of its labels.

**Data Augmentation.** We implement data augmentation by rotating the scene point cloud and grasping labels around the camera axis with a random angle uniformly sampled from the interval $[0, 2\pi)$. No further augmentations are needed.

**Ground-truth Graspness Definition.** For each training scene, we define a graspness score for the surface points of each object to represent its graspability. This score is determined by identifying a seed point and then assigning graspness to the nearby points. For an object $o$ in this scene, we denote all valid grasp labels that target $o$ as $G_o = \{g_o^i\}$, and the surface points of $o$ as $P_o = \{p_o^j\}$. We then define a grasp cone with $c$ being the apex, vector $cm$ being the axis and an aperture of $60°$, as shown in Fig. 14. Subsequently, we compute the projected distance of vector $cp_o^j$ along $cm$, denoted as $d$, and the spanning angle $\theta$ between $cp_o^j$ and $cm$. Using these quantities, the value of $f(g_o^i, p_o^j)$ is defined in Eq. 1. Numerically, this function is designed to attenuate exponentially with response to $\theta$ and $d$, halving at $10°$ or 1.5 cm. Then the seed point is defined as the point with the largest $f$ as shown in Eq. 2.

Finally, the seed point assigns graspness to nearby points with exponential decay and the graspness score of $p_o^j$ is computed as the logarithm of the sum of all contributed graspness, as in Eq. 4. Empirically, this score reflects the number of valid grasp labels near $p_o^j$.

From another perspective, this correspondence implicitly defines a grasp distribution conditioned on a point within a scene. Although articulating this distribution in precise mathematical terms is difficult, we contend that it objectively exists. This distribution represents the target distribution that the grasp generation module approximates.

Figure 14: Grasp cone for the graspness definition.

$$f(g_o^i, p_o^j) = \begin{cases} 0 & p_o^j \notin \text{this cone} \\ \exp(-\frac{\ln 2}{10}\frac{180}{\pi}\theta - \frac{\ln 2}{0.015}d) & p_o^j \in \text{this cone} \end{cases} \quad (1)$$

$$\text{seed\_point}(g_o^i) = \arg\max_{p_o^j \in P_o} f(g_o^i, p_o^j) \quad (2)$$

$$h(g_o^i, p_o^j) = 10^{-150\|\text{seed\_point}(g_o^i) - p_o^j\|_2} \quad (3)$$

$$\text{graspness\_score}(p_o^j) = \ln\left(0.001 + \sum_{g_o^i \in G_o} h(g_o^i, p_o^j)\right) \quad (4)$$

## E.2 Model

**Network Structure.** In the following paragraph, we elaborate on the network structures of our feature extractor, denoising model, graspness MLP, and joint MLP. First, our feature extractor employs the ResUNet14 architecture implemented with MinkowskiEngine [50] to derive point-wise feature vectors $f_p \in \mathbb{R}^{512}$ from a scene point cloud $P$, which is quantized into sparse voxels. This network resembles the one utilized in GSNet [18]. Second, our denoising model $v_\Theta(\hat{g}_E^t, f_s, t)$ is implemented as an MLP with layer sizes (524, 512, 256, 12) and Mish activations [51]. This model embeds $t$ into $\mathbb{R}^{512}$ using sinusoidal position embedding, adds this embedding with $f_s$, concatenates the resulting sum with $\hat{g}_E^t$, and feeds this concatenation into the MLP to predict the velocity. Third, our graspness MLP comprises a single-layer linear transformation, which maps $f_p$ to three values. The first two are interpreted as binary classification logits indicating whether this point is an object point, while the third value represents the predicted graspness score $GP_p$. Fourth, our joint MLP is a 6-layered MLP with ReLU activations and residual block designs following [52].

**Detailed Diffusion Dynamics.** The forward and backward processes of the diffusion each consist of $T_{\text{train}}$ and $T_{\text{inference}}$ time steps, respectively, evenly distributed within the interval $[0, 1]$. Additionally, the number of time steps of the backward process is required to be a divisor of that of the forward process. We denote the interval between two neighboring time steps of the backward process as $dt = 1/T_{\text{inference}}$. The DDPM [34] scheduler is employed to schedule the forward process variances $\beta_t$ for each time step $t = i/T_{\text{train}}, i = 1, 2, \ldots, T_{\text{train}}$:

$$\beta_t = \beta_{\min} + \frac{i-1}{T_{\text{train}} - 1}(\beta_{\max} - \beta_{\min}) \quad (5)$$

where $\beta_{\min}, \beta_{\max}$ are hyper-parameters. Then we define $\alpha_t = 1 - \beta_t$ and its cumulative product as $\overline{\alpha}_t = \prod_{j=1}^i \alpha_{j/T_{\text{train}}}$. At each training step, $\overline{\alpha}_t$ is utilized to determine the magnitude of noise to be added to the sample, as detailed in the main paper. At each inference step, we denoise a noisy sample $\hat{g}_E^t$ into a less noisy sample $\hat{g}_E^{t-dt}$ by solving the following ODE with $t$ from 1 to 0:

$$\hat{g}_E^t - \hat{g}_E^{t-dt} = d\hat{g}_E^t = \frac{T_{\text{train}}\beta_t\sqrt{\overline{\alpha}_t}}{2\sqrt{1 - \overline{\alpha}_t}}v_\Theta(\hat{g}_E^t, f_s, t)dt \quad (6)$$

Moreover, [38, 36] introduce a PDE to estimate the probability $p(g_E|f_s)$:

$$\frac{\partial \log p(\hat{g}_E^t|f_s)}{\partial t} = -\text{Tr}\left(\frac{\partial \overline{v}_t}{\partial \hat{g}_E^t}\right), \quad \text{where } \overline{v}_t = \frac{T_{\text{train}}\beta_t\sqrt{\overline{\alpha}_t}}{2\sqrt{1 - \overline{\alpha}_t}}v_\Theta(\hat{g}_E^t, f_s, t) \quad (7)$$

Based on the above equation, we can approximate a sample's probability $p(g_E|f_s)$ with numerical integration during the backward process. We rank each output $g$ of the grasp generation module using a linear combination of the estimated probability $p(g_E|f_s)$ of the wrist pose $g_E$ and the predicted graspness $GS_s$ of the seed point $s$:

$$\text{rank}(g) = p(g_E|f_s) + \eta GS_s \quad (8)$$

| Hyper-parameter | Value | Hyper-parameter | Value | Hyper-parameter | Value |
|---|---|---|---|---|---|
| Scene in each Batch | 8 | Grasp in each Scene | 64 | Init LR | 1e-3 |
| LR Scheduler | Cosine | Iter | 50000 | Point Num | 40000 |
| Voxel side length | 0.005 m | $k_{\text{trans}}$ | 25 | $T_{\text{train}}$ | 1000 |
| $T_{\text{inference}}$ | 200 | $\beta_{\min}$ | 0.0001 | $\beta_{\min}$ | 0.02 |
| $\lambda_o$ | 1 | $\lambda_g$ | 1 | $\lambda_d$ | 10 |
| $\lambda_\theta$ | 1 | $\eta$ | 10 | | |

Table 11: Hyper-parameter Setup

**Inference Speed and Memory Cost.** Our model efficiently processes a scene point cloud comprising 40,000 points, generating 128 grasp poses and ranking them all within **0.5 seconds**. The maximum memory usage during this inference is approximately **3 GB**. These evaluations were conducted on an NVIDIA 4090 graphics card.

# F  Implementation Details for Parallel Grippers

## F.1  Data Filtering and Refinement

As our generative model considers all grasping poses from the dataset as successful, and since the original GraspNet-1Billion dataset [14] includes some imperfect poses, we introduce a data filtering and refinement process before training. We retain only the grasping poses with a score of $\geq 0.9$ to ensure that all can successfully grasp the object with a friction coefficient of 0.2. To simplify motion planning, we assume that all grasps can be achieved by moving along the approaching vector and filtering out poses that would result in collisions during this movement. We also fix the depth to 4 cm and adjust the translation accordingly.

To handle poses that collide with the object and the table, we calculate the upper ($u$) and lower ($l$) bounds of the distance between the fingers along the original approaching vector. If the distance between any finger and the object is $u - l < 1.5$ cm, we discard the pose. We then uniformly sample new finger positions from the adjusted lower bound $l' = l + s$ and the adjusted upper bound $u' = l' + \min(0.01, (u - l - 0.01) - 2s)$, where $s = \min(0.01, \frac{u-l-0.01}{2})$. This ensures the fingers maintain a safe distance from the object without being too far. Finally, we calculate the intersection point of the object mesh and the new approaching vector, setting it as the seed point. Poses without a valid seed point are filtered out.

## F.2  Graspness Definition for Gripper

For parallel grippers, after we define the intersection point as the seed point, we assign the graspness to nearby points with Eq. 3 and compute the total graspness for each point with Eq. 4, same as the dexterous hand experiments.

## F.3  Sampling Poses from Prediction

Given the variability in graspness among different objects, we developed a new sampling strategy to maintain diversity and select high-quality grasping poses. First, we identify all seed points within the top 1% for graspability. For each of these seed points, we collect all points within a 2 cm radius. We then select the top 10% of these points based on graspability as new seed points and calculate grasping poses with them.

## F.4  Real-World Experiments

As a lot of the objects in the LEAP Hand's experiment are too large for our parallel gripper, we use different scenes in those two experiments as shown in Fig. 10.

# G   Additional Visualizations

In Fig. 15 we present more scenes with the predictions of our network. All point clouds are colored with a heatmap of model predicted graspness, with lighter color meaning higher graspness. Each scene is also dubbed with the predicted grasping pose corresponding to the highest rank.

In Fig. 16 we show some renderings of test scenes composed of objects from ShapeNet [31].

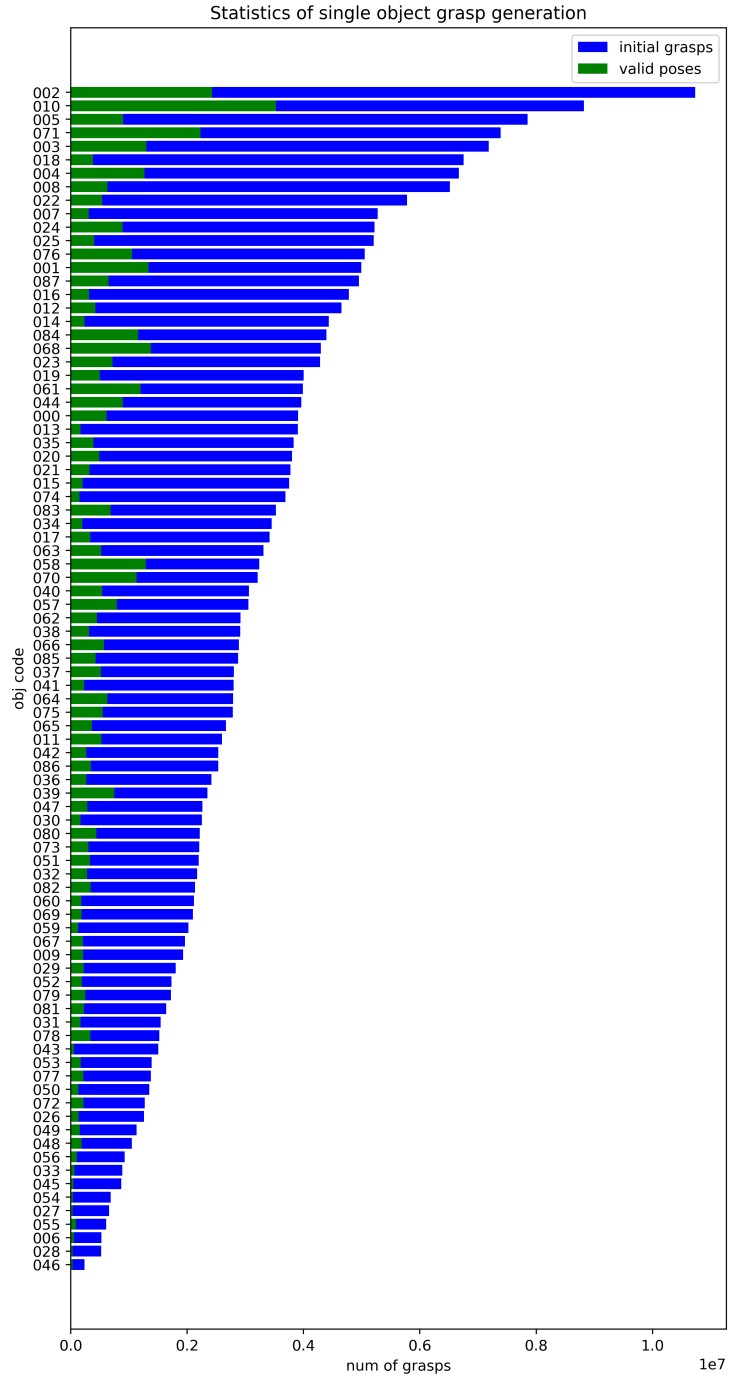

Figure 11: **Number of per-object initial grasp poses**. The proportion corresponding to valid grasps after optimization is colored green.

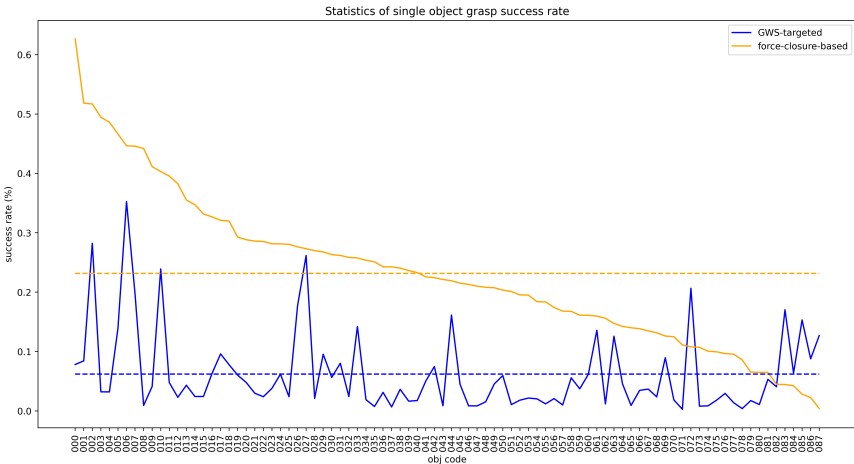

Figure 12: **Valid Rate** of single object grasp synthesis in sorted order. **Yellow** and **Blue** curves present per-object valid rates for our force-closure-based optimization method (Sec.D.2.2) and GWS-based optimization method (Sec.D.2.1), respectively. Averaged success rates are drawn in dotted lines, with values 24.19% and 7.91% respectively.

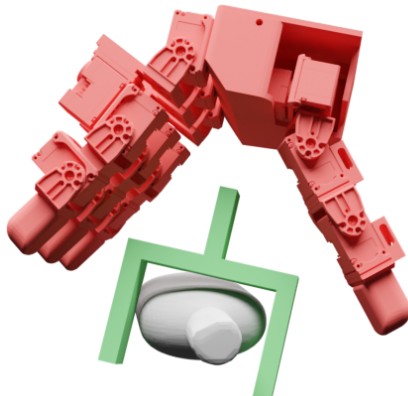

Figure 13: **Initial dexterous hand pose** superimposed with gripper grasp label at the same grasp point. We retarget gripper annotation in GraspNet-1Billion [14] to the initial 6D wrist pose of the dexterous hand and use a predefined set of joint qpos for initialization.

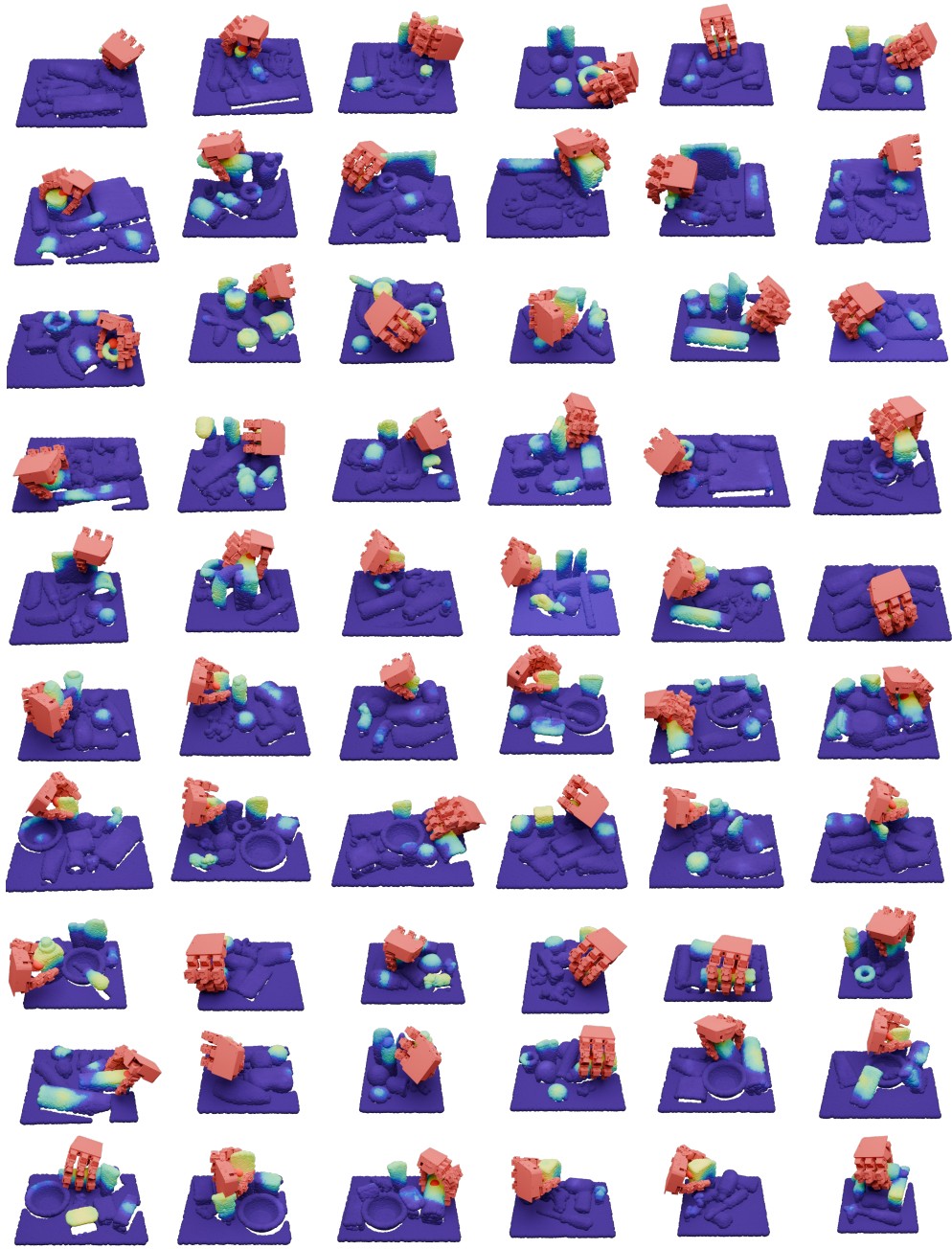

Figure 15: **Gallery visualization** of test scenes in our benchmark, corresponding to scenes 0100-0159 in GraspNet-1Billion [14]. All point clouds are colored with a heatmap of model predicted graspness, with lighter color meaning higher graspness. Each scene is also dubbed with the predicted grasping pose corresponding to the highest rank.

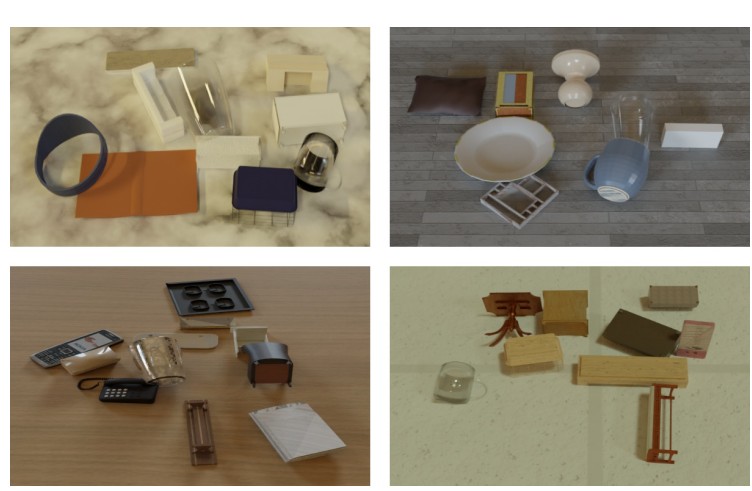

Figure 16: Test scenes composed of objects from ShapeNet [31].

