# OpenReview forum: "DexGraspNet 2.0: Learning Generative Dexterous Grasping in Large-scale Synthetic Cluttered Scenes"
_robot-learning.org/CoRL/2024/Conference — CoRL 2024_

### Official Review · Reviewer_2mB6 · 2024-07-19
**Potentially important contributions, but many open questions**

**Originality:** 3
**Technical Quality:** 4
**Clarity Of Presentation:** 3
**Potential Impact:** 3
**Recommendation:** 3
**Confidence:** 4

**Review:**

## Strengths

The paper proposes a large dataset for dexterous grasping in clutter, which seems to be a first attempt at this specific niche. There is a decently large variety of scenes in the dataset, and the authors explain how more scenes can be generated relatively easily. The dataset structure is explained clearly, and if released, it seems like it will be easy to use.

Considering the grasp generation pipeline, I like the focus on local features as a means of targeting generalization - it is evident from the ablation experiment that these allow for significant improvements in grasp generation. In addition, the choice of diffusion models for grasp generation is a benefit, as the distribution of grasp poses can easily be complex and multi-modal.


## Weaknesses

1 - In general, I am not sure about the intended focus of the paper. If the focus is the dataset itself (as suggested by the title and the abstract) I’d expect more details in section 3, explaining the design choices of generating the dataset and focusing on ablations relating to the dataset itself in the experiment section. If the focus is the grasp generation method, then I’d expect that to be the focus of the title, abstract and introduction as well.

2 - It seems like the grasp generation method simply selects a single grasp for the entire clutter of objects - there is no way to distinguish between objects (only between objects and background) and no way to select which object to grasp. Is this the case, or am I misunderstanding the object segmentation score? If it is, this is a limitation of the method that needs to be mentioned. This could probably be alleviated by incorporating a point cloud segmentation module for preprocessing; however, it might require significant changes in the architecture or training pipeline.

3 - In general (following up on the previous point), the limitations section is non-existent - a single sentence in the conclusion paragraph states things that can be added, but not actual limitations of this approach. What scenes do the authors expect this method to fail on? How much wall time does grasp generation take? etc.

4 - The Related Work section is lacking.
First, it can use a general polish - the writing is unclear at places, which makes it hard to understand how each paper mentioned is relevant as related work. Some examples: what are “graspness” (line 60) and “data confusion” (line 61)? “… diversity and quality are both limited with an easy setting” (line 74) what is the setting?

In addition, some papers are missing from the data-driven dexterous grasping subsection. A few examples:

Data-driven dexterous grasping:
- [Dexterous Functional Grasping](https://arxiv.org/abs/2312.02975), Agrawal et al., CoRL 2023 : Uses training in simulation to generate dexterous grasps with LEAP hand

Generative models for grasping:
- [6-DOF GraspNet: Variational Grasp Generation for Object Manipulation](https://arxiv.org/abs/1905.10520), Mousavian et al., ICCV 2019 Uses VAE as generative model for grasp generation
- [Increasing the Generalisation Capacity of Conditional VAEs](https://arxiv.org/abs/1908.08750), Klushyn et al., ICANN 2019

Grasping in clutter:
- [Contact-GraspNet: Efficient 6-DoF Grasp Generation in Cluttered Scenes](https://arxiv.org/abs/2103.14127), Sundermeyer et al., ICRA 2021
- [Multi-View Picking: Next-best-view Reaching for Improved Grasping in Clutter](https://arxiv.org/abs/1809.08564), Morrision et al., ICRA 2019. While their approach is mentioned in the baselines, this paper is not discussed in the related work section

Datasets for Grasping:
- [MetaGraspNetV2: All-in-One Dataset Enabling Fast and Reliable Robotic Bin Picking via Object Relationship Reasoning and Dexterous Grasping](https://ieeexplore.ieee.org/document/10309974), Gilles et al., IEEE TASE 2023. While not providing dexterous grasp examples, this dataset contains a large set of cluttered scenes in simulation.
- [ACRONYM](https://sites.google.com/nvidia.com/graspdataset), Eppner et al., ICRA 2021, Large dataset of grasp points based in simulation.


### Typos and minor errors:

Line 44, last paragraph of the introduction, “DexGrasNet”

Lines 99, 107 and others: “supp” I assume means “supplementary material”?

Line 140 “object segmentation score $GS_p$” instead of $O$

Line 153 “multi-moded” → “multi-modal”?

Line 256 “Fig.2” → “Table 2”

**Quality Of The Limitations Section:**

1

**Questions For Rebuttal:**

1. Why use only 60 objects for training and keep >1200 other objects for testing only? Seems like much more benefit could be unlocked by using additional objects and scenes.
2. How are scenes physically constructed in simulation? When objects are deleted from “seminal” scenes, is physics allowed to run for the other objects to rearrange themselves?
3. Line 133 in Graspness Definition: what heuristic rule was used for selecting the nearby grasp points? Were any ablations run on this heuristic (controlling, in essence, the local region size and shape)? Also in line 142, why does FPS make sense? Why not NMS on graspness score?
4. Why are success rates on the Loose scenes lower? I’d expect grasping to be easier with less clutter.
5. Grasp pose success is evaluated by the ability to pick up an object in simulation. However, this depends on the contact properties of the physics simulation (friction, collision mesh inaccuracies etc.). Have the authors investigated the relevant properties in the simulator to find the best parameters for sim2real transfer?
6. How long does the pose diffusion process take? In particular, for the real world grasps - is this process a realistic option for real-time grasping?

**Robotics Focus:**

4

**Summary Of Paper:**

This paper presents a dataset of cluttered scenes and grasp poses in simulation for the dexterous LEAP hand, consisting of a large number of scenes for a variety of objects, with a very large number of labeled grasp poses. In addition, a grasp pose generation pipeline based on diffusion models is proposed and evaluated on the new dataset.

**Summary Of Recommendation:**

In general this has valid contributions and is potentially a good match for CoRL, however some questions and concerns remain that prevent it from being accepted as is. ** Post-rebuttal update: As most of my questions and concerns were addressed, I am happy to raise my score accordingly. **

---

### Official Review · Reviewer_bSuk · 2024-07-20
**Large scale grasping dataset with generative model for dexterous grasping**

**Originality:** 4
**Technical Quality:** 4
**Clarity Of Presentation:** 4
**Potential Impact:** 3
**Recommendation:** 3
**Confidence:** 5

**Review:**

# Strengths:

- The paper's key contribution is in the large scale grasping dataset with lots of diversity in object and cluttered scene variations. This allows the authors to systematically analyse how much data scaling helps in performance of grasping models and what factors are actually important (scene diversity v/s number of object count).

- Another highlight of the paper is the decomposed pose and joint angle prediction for which a clear reason is mentioned in terms of differences in data distributions for pose and joint angles. However it would be good to move this discussion from supplementary to the main paper to motivate this reason more. Although the overall generative framework is not exactly novel, the authors provide sufficient explanation and reasoning for these aspects.

- I especially liked the depth of ablation experiments and details provided in the supplementary material wrt the experiment and modeling setup. The proposed method shows clear improvements (in simulation) over baseline methods across all object categories though it remains to be seen if the inference runtime costs is prohibitive.

# Weakness:

- The discussion of failure cases seems to be limited. In general, harder to grasp objects will have a low success rate but the readers will benefit from an author's view about where their method struggles and if any simple refinements were used for the experiment trials.

- The real-world experiments also seem to be limited both in the number of scenes tested out and in the baseline methods compared against. It was also not clear from the main paper as to why an additional method for test-time depth is needed and how the performance of real-world experiments is affected due to this factor?

- It is also not clear if the dataset generation scheme is scalable to many different grippers which might have different kinematics. It would be good to mention the relative effort needed to port such a dataset to new grippers.

# Suggestions:

- Possible typo at Line-#228: text says 75.4% success rate at 1/1000 scale whereas in the figure its shown as 74.5%

**Quality Of The Limitations Section:**

2

**Questions For Rebuttal:**

- Please provide some analysis or discussion about runtime costs for the real world pipeline especially with test-time depth restoration as an augmentation method.

- Was there any particular reason to not test other grasping methods in the real world experiments (eg. gripper limitation, unseen object set etc.)

**Robotics Focus:**

4

**Summary Of Paper:**

This paper tackles the problem of generating valid grasps in a cluttered scene for dexterous hands (Leap gripper). The paper's key contribution lies in the large-scale dataset of cluttered scenes with valid grasps along with an analysis on different factors affecting grasping success. Along with the dataset, a grasp generation module is proposed where the pose and joint configuration are predicted in a conditional framework and shown to have better results than direct joint modeling. Experimental results show a clear increase in performance when compared with previous generative/regression models along with showing that diversity in training scenes is more important for scaling up the performance of grasping.

**Summary Of Recommendation:**

The paper seems to contribute with a large-scale grasping dataset and also shows some scaling experiments which would be the highlight to a reader. The proposed  generative model is not exactly new but the authors do seem to reason about their design choices and show a general improvement over related works. A potential drawback is the lack of sufficient real-world testing in number of scenes.

---

### Official Review · Reviewer_AfH6 · 2024-07-23
**Promising Paper on Generating Dexterous Grasps**

**Originality:** 4
**Technical Quality:** 4
**Clarity Of Presentation:** 4
**Potential Impact:** 3
**Recommendation:** 3
**Confidence:** 4

**Review:**

The paper predicts grasps in multiple stages: First it segments grasp points, after that it uses local features for each grasp point to predict R, T of the wrist using a diffusion model and finally predict the hand joints using a MLP. The backbone of the model is ResUNet14 built on top of MinkowskiEngine. The model is trained on synthetic data that is one of the contributions of the paper. The dataset is generated in isaac gym.

Strengths:
- The paper performs quite well in the real world. Even when there is partial depth from transparent objects which is quite surprising.
- The dataset can be quite useful for further research in dexterous grasping. Specially given the scale of the dataset which contains 426M grasps.
- I like the additional ablation studies in the supplementary that already covered my questions on the effect of different scale of data and other choices for rotation prediction.

Rebuttal questions:
- In the supplementary video there is a part where it shows it can grasp transparent objects without any need for depth restoration or improvement. I would like to see the point cloud for these scenes to better understand the behavior of the model. It is really surprising for a method that only uses point cloud can work.

- Following up on the point, it would strengthen the paper if authors try scenes (Let's say 10 scenes) with only transparent objects at different poses and report the performance on those objects in different scenes. It would clarify whether it is a coincidence or the method is indeed robust toward partial point cloud or even corrupted point cloud of the transparent objects.


Minor comment:
Line 95 ojbects -> objects.

**Quality Of The Limitations Section:**

2

**Questions For Rebuttal:**

See above.

**Robotics Focus:**

4

**Summary Of Paper:**

The paper learns a model to generate dexterous grasps from point cloud. It is trained on simulation data and achieves 91% success rate on the real experiments.

**Summary Of Recommendation:**

I think this is a good paper with impressive results. I am quite positive on the paper.

---

### Author Rebuttal · Authors · 2024-08-11

We thank all reviewers and the chairs for their valuable comments. We address all questions and concerns in the official comments below each review and the meta review, and will incorporate all feedback into the revision.

In the rebuttal zip file, we provide the RGB image, raw point cloud, and restored point cloud of a scene with diffuse objects and a scene with transparent or specular objects for reviewer AfH6.

---

### Decision · Program_Chairs · 2024-09-04

**Decision:**

Accept

**Comment:**

Post-Rebuttal Meta Review:
-------------------------------------------
The paper presents an practical dataset for dexterous grasping. Given that the rebuttal and responses addressed the major concerns of the reviewers, I am happy to recommend the paper for acceptance. We look forward to the discussion and changes being reflected in the camera-ready version.

Original Meta Review:
-------------------------------------------
This paper proposes DexGraspNet2.0, a large-scale simulation-based benchmark for robotic grasping in clutter scenes.

Strengths: This paper presents a very large-scale dataset that can be built upon and leveraged by other researchers. The generation methodology employs interesting ideas, and the proposed method is shown to work well in the real world. A large strength of the paper, as noted by all reviewers, is the excellent systematic analysis and simulation ablations that highlight and justify the design choices.

Weakness: The paper does not currently sufficiently discuss limitations and the failure cases (Reviewer AfH6, Reviewer bSuk, R3). While the paper contains many valuable contributions, it would benefit from more explicitly stating and focusing these contributions (Reviewer 2mB6). There are several technical questions that need resolving: how the method handles transparent objects (Reviewer AfH6), how the method could generalize to different grippers (Reviewer bSuk), how the method distinguishes between objects (Reviewer 2mB6), and the runtime, particularly in the real-world system (Reviewer bSuk, Reviewer 2mB6). The related work section could be expanded upon and one reviewer provides detailed suggestions (Reviewer 2mB6).